# Foreign Body Aspiration in Children—Retrospective Study and Management Novelties

**DOI:** 10.3390/medicina59061113

**Published:** 2023-06-09

**Authors:** Dana Elena Mîndru, Gabriela Păduraru, Carmen Daniela Rusu, Elena Țarcă, Alice Nicoleta Azoicăi, Solange Tamara Roșu, Alexandrina-Ștefania Curpăn, Irina Mihaela Ciomaga Jitaru, Ioana Alexandra Pădureț, Alina Costina Luca

**Affiliations:** 1Department of Mother and Child Medicine, University of Medicine and Pharmacy ”Gr.T.Popa”, 700115 Iasi, Romania; 2Clinical Hospital of Emergency for Children Sfanta Maria, 700309 Iasi, Romania; 3Department of Biology, Faculty of Biology, “Alexandru Ioan Cuza” University of Iasi, 700505 Iasi, Romania

**Keywords:** pediatric emergency, foreign body aspirations, bronchoscopy, algorithms

## Abstract

Foreign body aspiration (FBA) is a frequent diagnosis in children. In the absence of other lung conditions, such as asthma or chronic pulmonary infections, this manifests as a sudden onset of cough, dyspnea, and wheezing. The differential diagnosis is based on a scoring system which takes into account the clinical picture as well as the radiologic aspects. The treatment that is considered the gold-standard for FBA in children remains to be rigid fibronchoscopy, however it comes with several potentially crucial local complications such as airway edema, bleeding, and bronchospasm, along inherent issues due to general anesthesia. *Material and methods*: Our study is a retrospective study analyzing the medical files of the cases from our hospital over the span of 9 years. The study group consisted of 242 patients aged 0–16 years diagnosed with foreign body aspiration in the Emergency Clinical Hospital for Children “Sfânta Maria” Iași, between January 2010–January 2018. Clinical and imaging data were extracted from the patients’ observation sheets. *Results:* In our cohort, the distribution of children with foreign body aspiration was uneven, with the highest incidence being reported in children from rural areas (70% of cases), whereas the most affected age group was 1–3 years, amounting to 79% of all cases. The main symptoms which led to emergency admittance were coughing (33%) and dyspnea (22%). The most important factors that determined the unequal distribution were socio-economic status, which relates to a lack of adequate supervision by parents, as well as the consumption of food inappropriate for their age. *Conclusions:* Foreign body aspiration is a major medical emergency that may be associated with dramatic clinical manifestations. Several scoring algorithms designed to establish the need for bronchoscopy have been proposed, taking into account both the clinical and radiological results. The issue with asymptomatic or mild symptomatic cases, as well as difficulties managing cases with radiolucent foreign bodies, remains a challenge.

## 1. Introduction

Tracheobronchial foreign body aspiration (FBA) is a major problem in pediatrics, and is a significant cause of accidental fatalities among children under the age of three [1,2]. The diagnosis is often omitted initially, especially in children with an unknown history and difficulties in performing anamnesis. In around 30% of cases, the symptoms are treated as non-specific conditions [3]. FBA may lead to chronic health problems (including a chronic cough, bronchiectasis, and recurrent pneumonia) [4] and irreversible lung injuries, or even become life-threatening by leading to cardiopulmonary arrest and sudden death [2]. FBA is more common in infants and toddlers due to the lack of a complete denture, the tendency to bring various objects to their mouth, moving while eating, and having a weak protective laryngeal reflex [5,6]. All of these aspects can lead to objects or particles getting stuck in the glottal opening, larynx, or trachea [7]. At a young age, the most common foreign bodies that are organic include peanuts and seeds, which then progressively shifts towards inorganic materials (including coins, paper clips, and pins) with age [8]. The shape of the foreign body can act an indicator for its final destination in the airway, as slimmer, sharper objects tend to travel further down the bronchial tree, becoming extremely dangerous if they reach the cricoid membrane [9]. Most foreign bodies pass through the larynx and the trachea, followed by the left and right bronchial tree [10], and in some cases the foreign bodies reach the stomach and the intestines. In this case, larger objects are able to cause pyloric obstruction accompanied by vomiting and eating refusal [11]. However, in most cases foreign bodies reaching the esophagus, stomach, and intestines are asymptomatic, and transverse the gastrointestinal tract without issues [12].

The identifying symptoms (sudden cough and wheezing) are not always present as they are dependent on a series of factors such as the type of the object, age of the child, the time passed since the event, and the degree of airway blockage [8,13], all of which can cause a delayed display of respiratory symptoms [14]. Although chest radiography is generally the first chosen diagnosis option, it may give false-negative results in up to 30% of cases as most foreign bodies are not radiopaque. Whereas indirect signs such as emphysema, atelectasis, and pulmonary infiltrate are not present in all patients [15,16], rigid bronchoscopy under general anesthesia [17] therefore becomes essential as it serves a dual purpose—diagnosis and therapeutic treatment if the diagnosis is confirmed. However, its influence on the survival rate should always be taken into account [18]. FBA used to be one of the most common causes of death for children under the age of three, but this aspect has significantly decreased due to the significant medical advancements in the field of bronchoscopy techniques [19].

The present study was performed with the following objectives: 1. Analysis of the relationship between the type of aspirated foreign body and the patient’s age; 2. Symptoms and complications; 3. Patients classification based on the clinical forms of the disease; 4. The evaluation of significant predictors of foreign body aspiration based on anamnesis, clinical, and radiological examinations; 5. The results of bronchoscopy in patients suspected of foreign bodies aspirated into the respiratory tract; 6. The therapeutic methods of foreign body aspiration.

## 2. Material and Methods

The present study is retrospective analytical study, and follows the STROBE criteria [20]. The “Sfanta Maria” Emergency Clinical Hospital is a tertiary hospital. For the purposes of this paper, we have used the observation files from the archives and included the patients from this hospital that met all the inclusion criteria (age between 0 and 16 years old, a hospitalization period greater than 24 h, bronchoscopy performed during hospitalization, and confirmed diagnosis of foreign body aspiration). The study was conducted in accordance with the Declaration of Helsinki and Romanian Law no. 206/27.05.2004, and approved by the Ethics Committee of ”Grigore T. Popa” University of Medicina and Pharmacy (no. 36915/20.12.2022).

We included in the study a total of 232 patients with ages between 0–16 years, diagnosed with foreign body aspiration in the “Sfanta Maria” Emergency Clinical Hospital for Children, Iași between January 2010–January 2018. The following data were extracted from the patient observation charts: age, sex, place of origin, admission symptoms, the time elapsed from inhalation to diagnosis, nature of the aspirated foreign body, location in the tracheobronchial tree, pulmonary clinical examination, results of imaging investigations, bronchoscopy results, complications, antibiotic therapy, and days of hospitalization.

A child who presents with a sudden cough and wheezing in the absence of any disease such as asthma or chronic lung infections may be suspected of having foreign body aspiration with symptoms and severity depending on the time elapsed between the inhalation of the foreign body and hospital admission (during or after the first 7 days in the case of our study). The diagnostic method was either direct visualization of the foreign body if possible or bronchoscopy in most cases. Potential confounders included an age younger than 3-years-old due to the natural exploration tendencies, the hand-to-mouth coordination, as well as the need for much stricter parental supervision, which is not always perfect, especially in dysfunctional families as presented in our study.

We have performed a descriptive statistical analysis of the aforementioned qualitative and quantitative variables by means of a standard statistical package (JASP TEAM 2022, Version 0.16.1, University of Amsterdam, The Netherlands).

## 3. Results

We only included and analyzed the patients that met the criteria with no individual having being dropped or excluded from the selected group. There was no need for follow-up, as most of our patients were kept under observation until they made a full recovery, except for those who were discharged on request against medical advice.

The male: female ratio was 1.77. Of the confirmed cases, 17 were under the age of 1-year-old (7%), 182 cases between 1–3 years old (79%), and 33 cases aged between 4–16 years old (14%), respectively. There was an increased incidence between 1–3 years old, followed by the 4–16 years interval. Patients younger than 1-year-old were rarely identified in our study group. This case distribution can be explained on one hand by the fact that children between 1–3 years old tend to explore the environment, and thus can aspirate different foods or objects, and on the other hand, by the lack of adequate parental supervision. Furthermore, 69.83% of patients came from rural areas, including from dysfunctional families, where poverty, a lack of health education, divorces, alcohol consumption, and toxic living environments have made their impact based on the medical files available and the family history. Comparing the socio-economic characteristics of our group with those of the groups reported in other studies, we considered that the precarious economic and educational status seems to be a key characteristic in our area. This led to the logical conclusion that part of the prophylaxis of foreign body aspiration is related to educating the population and applying the measures that are destined to increase the living standards through implementing the appropriate policies.

In our cohort, there was a higher incidence of organic (vegetable) foreign bodies (FBs) (192 patients) than inorganic ones (20 patients). We thus deduced that the main responsible factor is the consumption of inappropriate food.

The highest incidence of vegetable bodies was identified in the 1–3 years old age group (86%), and that of mechanical objects in the 4–16 years old age group (50%), respectively. The lowest incidence of cases, both organic and inorganic, were identified in patients under 1-year-old.

The classification of organic foreign bodies according to their incidence in the studied group revealed a much higher frequency of seed aspiration: 5% of cases (111 children), compared to hazelnut aspiration 23% of cases (47 children), walnut kernel 13% of cases (26 patients), corn kernel in 5% of cases (10 cases), and beans 4% of patients (8 cases). Seed aspiration is a particularly complex clinical situation, as they contain volatile oils that can cause rapid bronchial damage, along with the volume expansion of the seeds in the bronchus due to imbibition that can lead to the complete obstruction of the airways.

Twenty-one percent of patients were admitted within the first 24 h with symptoms of foreign body aspiration or a significant anamnesis, whereas 16.94% were admitted with post-inhalation complications 7 days after the event.

The main clinical manifestations were as follows: sudden onset of cough (33.05%), dyspnea (21.9%), acute respiratory failure (19%), wheezing (14.87%), and cyanosis (9.09%). One of the causes for a delayed diagnosis was the minimal symptoms that were encountered in several cases. In our cohort, this was the case for 2% of all patients.

The amount of time elapsed between the inhalation of FBs and emergency significantly affects the clinical picture. For patients admitted during the first 7 days after the aspiration, cough was the most common symptom (33%), followed by wheezing (24%), and acute respiratory failure (18%) (Figure 1).

After the first 7 days (henceforth termed as late-admission), respiratory symptoms worsened, and the main reason for emergency room visits in 42% of our patients was dyspnea therefore falling under the late-admission category.

The initial clinical examination revealed decreased lung sounds in 78 cases, bronchial obstruction and coarse vesicular murmur in 55 cases, and bronchial rales in 116 patients, while suggestive aspects for aspiration pneumonia and pulmonary obstruction were noted in 32 children, respectively.

The most common radiological feature revealed through chest radiographies (Figure 2) was pulmonary atelectasis (26.85%) followed by pulmonary emphysema (17.76%). The high incidence of atelectasis can be explained by the fact that 61.98% of patients visited a doctor more than 24 h after the aspiration event, thereby indicating that the timing of the diagnosis heavily influences the obstruction of the respiratory tract.

As direct visualization of the aspirated foreign body was only possible in eight cases, we opted to use rigid bronchoscopy as both a diagnostic and treatment tool (Figure 3). Regardless of the age group and sex, the most common location of foreign bodies was found to be the right bronchial tree (51.23%), followed by the left primitive bronchus (33.45%), trachea (7.02%), and tracheal bifurcation (5.37%). The upper airways were only involved in seven cases (nostrils and glottis), which could be explained by the fact that most of our patients (61.63%) came into the ER after the 24h framework, by which time the foreign body would have migrated.

Complications of foreign body aspiration in children are highly dependent on the time elapsed between inhalation and extraction. In our cohort, the most common issue was aspiration pneumonia (123 cases), followed by acute tracheobronchitis (13 cases), laryngitis in three cases, and lung abscess in two cases. Furthermore, five children required assisted ventilation, and one case required temporary tracheostomy.

Delaying treatment can increase the severity of complications. In this study, 84% of patients developed infectious lung diseases after an inhalation accident, whereas 88% of cases received antibiotic treatment either with curative or preventive purposes throughout the hospitalization period. In total, 204 out of 232 of our patients received antibiotic therapy. Furthermore, 54% of our cases had a hospitalization period longer than 5 days, whereas the rest were either hospitalized for less or were discharged on request against medical advice (9%). A summary of our results alongside observation chart data is illustrated in Table 1.

## 4. Discussions

The study of Tang LF et al. on a total of 1027 children with FBA reported a frequency of 82.6% cases in the 1–3 years old age group [21,22]. In another recent study of 316 children who underwent bronchoscopy, 69.9% were found between 1 and 3 years old [3,21]. Another interesting complex study is the one published by Ulas et al., 2022, who reported similar results to our findings, with the greatest number of cases identified in children younger than 3 years old, and the most important symptoms observed being a sudden cough and wheezing [1]. Meta-analyzes available in the literature also supports the fact that in the majority of cases, the foreign body is of an organic nature [23].

Available data from the literature shows a net predominance in males assessed of approximately 2/1. Thus, Tomaske M et al. reported that out of a group of 370 children with aspiration of foreign bodies, 242 (65.4%) were in males [15,24]. In another recent study, 67.9% of patients admitted for FBA were boys [25,26]. Another large group study (involving 1027 patients) reported their FB prevalence to be 626 boys to 401 girls [16,22]. Our findings are consistent with those previously reported, with a male-to-female ratio of 1.77.

The current management of FBA consists of bronchoscopy. However, several studies have reported negative results with important associated complications, rendering the need for clinical scoring systems that may reduce the number of unnecessary bronchoscopy examinations.

Haller et al. identified seven criteria that were found to be positively correlated with FB aspiration, which are as follows: sudden choking, apnea, and decreased lung sounds as clinical parameters, and lung atelectasis, air trapping, and mediastinal shift as radiological parameters [27]. Although not applied in the case of our study, the model of Haller for the diagnosis of foreign body aspiration with an accuracy of 91% stated that having any three of the following symptoms: sudden choking, cyanosis, apnea, decreased breath sounds, atelectasis, mediastinal shift, and air trapping are a clear indicator of foreign body presence. In our cohort, sudden choking was inconsistently reported, probably due to a lack of awareness. Decreased lung sounds and coughing were the most frequently identified clinical aspects, with positive predictive values being found to be consistent with those reported in other similar studies. Furthermore, we encountered the same radiological findings suggested by Haller et al., which thereby led us to conclude that the scoring system developed by Haller et al. could be of use, provided that future studies take into consideration the time elapsed from inhalation, as well as the underlying pulmonary conditions that may influence the clinical parameters [27].

Fasseeh et al. proposed another algorithm for the diagnosis of foreign body aspiration based on witnessed choking, sudden cough, new onset or recurrent wheezes, unilateral diminished breath sound, wheezy chest, and respiratory distress [28]. Coughing, wheezing, and lung sounds had the highest specificity, and were attributed three points each in their scoring system. A final score of five points or greater was used as an indication for bronchoscopy.

Generally, we have obtained similar clinical findings in our cohort compared to previous studies, and no negative bronchoscopy results were identified using the same clinical indicators as those suggested by Fasseeh et al. [28]. We indeed refined our management plan considering extensive anamnestic data and, in some cases, radiological findings, and found no cases of radiolucent foreign bodies interfering with the results. In their study, Fasseeh et al. found consolidation, bilateral hyperinflation, and lung infiltrates as highly specific for FBA [28]. It is, in our opinion, important to establish the amount of time it takes for these radiologic findings to become apparent. In our cohort, 55% of patients with FBA had no chest X-ray abnormalities at the time of their hospital admission, and this category included those who came in the first 24 h. Infectious lung complications were found in 17% of cases, all of which were admitted no later than 7 days after aspiration, leaving an important number of cases with clinical signs of aspiration and no radiologic findings.

It is also important to discuss the cases with radiolucent foreign bodies and minimal symptoms. Thoracic computer tomography (CT) has been proposed as a diagnostic tool for these cases, as well as for those with recurrent, treatment-resistant respiratory infections [29]. Due to radiation concerns, routine use of CT for the positive diagnosis of FBA is discouraged. However, in selected cases, it may be of use to establish an appropriate treatment plan. Ultrasonography is a useful alternative for evaluating the signs of pulmonary inflammatory syndrome and detecting the migration of FBs outside the respiratory airways [30,31].

Overall, our results are in accordance with the literature, with the observed differences mainly due to the countries/regions of the performed studies. However, in a future study we would consider applying a scoring system that has been shown to be useful in similar studies. Limitations of our study include the number of patients and the period of selection; however, we consider that this study also provided great examples of why education about this issue is of the utmost importance for the general public, especially for families with a lower socio-economic status where the strict supervision of babies and toddlers is lower.

## 5. Conclusions

Foreign body aspiration is still a life-threatening condition in infants and children. The advent of clinical and imagistic scoring systems has proven useful in evaluating the need for bronchoscopy and facilitating early treatment measures. There is still a need for the further validation of diagnostic algorithms, especially in cases presenting with discrete symptoms and normal chest radiography.

## Figures and Tables

**Figure 1 medicina-59-01113-f001:**
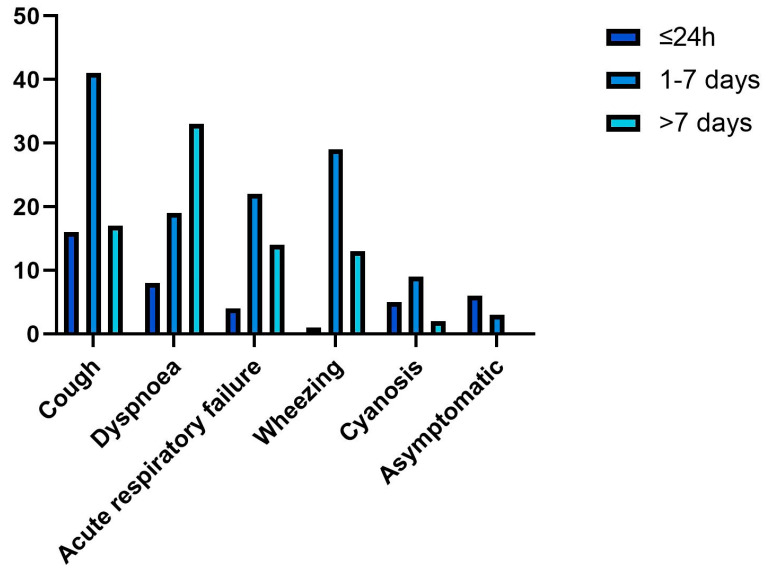
Distribution of symptoms based on the time elapsed between inhalation and medical examination.

**Figure 2 medicina-59-01113-f002:**
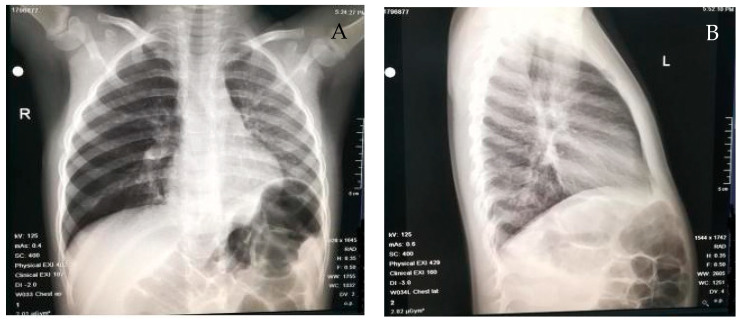
Examples of radiologic findings in our cohort: (**A**) the pulmonary transparency difference between the two pulmonary areas, particularly increased on the right (right lung emphysema), and (**B**) perihilar and left hiliobasal interstitial infiltrate which projects retrocardiacally.

**Figure 3 medicina-59-01113-f003:**
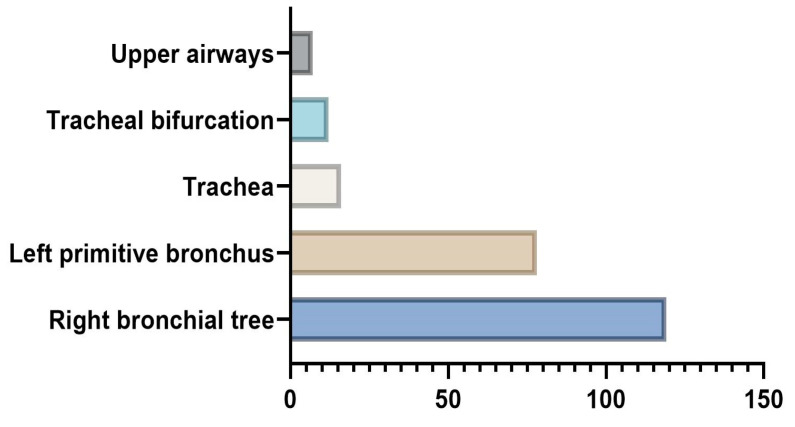
Anatomic location of foreign bodies, as identified with rigid bronchoscopy.

**Table 1 medicina-59-01113-t001:** Summary of the main results and observation chart data regarding 242 patients admitted with foreign body aspiration.

Characteristic	Frequency	Percentage
**Total number of patients**	232	
**Sex**		
Male:Female ratio	1.77	
**Age**		
<1 year old	17	7.32%
1–3 years old	182	78.45%
4–16 years old	33	14.22%
**Place of origin**		
Rural area	162	69.82%
Urban area	70	30.17%
**No. of cases based of the nature of the foreign body**		
Organic + inorganic bodies	10	4.31%
Organic (vegetable) bodies (alone and combined)	202	
Seeds	111	55%
Hazelnut	47	23%
Walnut kernel	26	13%
Corn kernel	10	5%
Beans	8	4%
Inorganic bodies	20	
**Time of admission**		
Within the first 24 h	49	21%
After the first 24 h	143	61.63%
After 7 days	40	16.94%
**Main clinical manifestation**		
Sudden onset of cough	77	33.05%
Dyspnea	49	21.09%
Acute respiratory failure	44	19%
Wheezing	35	14.87%
Cyanosis	21	9.09%
**Initial clinical examination**		
Decreased lung sounds	78	33.62%
Bronchial obstruction and coarse vesicular murmur	55	23.7%
Bronchial rales	116	50%
Aspiration pneumonia and pulmonary obstruction	32	13.79%
**Chest radiography results**		
Pulmonary atelectasis	62	26.85%
Pulmonary emphysema	41	17.76%
**Direct visualization of the foreign body**	8	3.44%
**Location of the foreign body**		
Right bronchial tree	124	51.23%
Left primitive bronchus	81	33.45%
Trachea	17	7.02%
Tracheal bifurcation	13	5.37%
Upper airways (nostrils and glottis)	7	3%
**Complications**		
Aspiration pneumonia	123	53.01%
Acute tracheobronchitis	13	5.6%
Laryngitis	3	1.29%
Lung abscess	2	0.86%
Assisted ventilation	5	2.15%
Temporary tracheostomy	1	0.43%
Antibiotic therapy	204	88%
**Period of hospitalization**		
<5 days	107	46%
>5 days	125	54%

## Data Availability

Data sharing not applicable.

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
