# Peer review of "Foreign Body Aspiration in Children—Retrospective Study and Management Novelties"

_medicina, 2023, doi:10.3390/medicina59061113_

Round 1
Reviewer 1 Report (New Reviewer)
Thank you for the opportunity to get acquainted with the interesting work. The authors describe the clinically important problem of foreign bodies in the respiratory tract in children. A foreign body in the respiratory tract of a child can be the cause of numerous diseases and even death of the child. In their work, authors should complete Table 1 with percentages. In their analysis, the authors should analyze whether the frequency of foreign bodies in the right and left main bronchus is statistically significantly different. It is puzzling why so few foreign bodies were found in the upper respiratory tract, whether the reason for this was their outpatient removal, please provide information and complete it.
I am trying to supplement the literature with:
Gvetadze P, Chkhaidze I, Baldas S, Comoretto R, Gregori D; Susy Safe Working Group. Injuries due to foreign body aspirations in Georgia: A prevention perspective. Int J Pediatrician Otorhinolaryngol. 2016 Apr;83:84-7. doi: 10.1016/j.ijporl.2015.12.007. Epub 2015 Dec 21.
Chmielik L.P., Fršckiewicz M., Chmielik M.: Diagnostic and therapeutic difficulties in cases of foreign body in the oesophagus in children treated in the Pediatric ENT Clinic WUM in 2005-2009. case reports. New Medicine. 2009;13(4):85-88.
Rance A, Mittaine M, Michelet M, Martin Blondel A, Labouret G. Delayed diagnosis of foreign body aspiration in children. Arch Pediatrician. 2022 Aug;29(6):424-428. doi: 10.1016/j.arcped.2022.05.006. Epub 2022 Jun 12.
There are many spelling and language mistakes and the manuscript needs to be corrected by a native English speaker.
Author Response
Dear Reviewer 1,
Thank your for your comments and suggestions. We have decided to answer as follows:
- In their work, authors should complete Table 1 with percentages
Thank your for your suggestion, we have completed the table with the percentages after verifying all data and numbers all over again. Thanks to you we have identified and corrected some mistakes that we have made while transferring data. We also replaced the graphs to suit better a proper scientific paper.
- In their analysis, the authors should analyze whether the frequency of foreign bodies in the right and left main bronchus is statistically significantly different.
We apologize, but to the best of our knowledge, there is no way to check whether two singular values are statistically different. If we misunderstood your request and you meant a different type of comparison, please let us know.
- It is puzzling why so few foreign bodies were found in the upper respiratory tract, whether the reason for this was their outpatient removal, please provide information and complete it.
We believe that a possible explanation for this would be the fact that most of our patients came into the ER after the 24h since the incident and the foreign body just migrated. We have added this statement into the main body text as well (Lines 202-204).
-
I am trying to supplement the literature with:
Gvetadze P, Chkhaidze I, Baldas S, Comoretto R, Gregori D; Susy Safe Working Group. Injuries due to foreign body aspirations in Georgia: A prevention perspective. Int J Pediatrician Otorhinolaryngol. 2016 Apr;83:84-7. doi: 10.1016/j.ijporl.2015.12.007. Epub 2015 Dec 21.
Chmielik L.P., Fršckiewicz M., Chmielik M.: Diagnostic and therapeutic difficulties in cases of foreign body in the oesophagus in children treated in the Pediatric ENT Clinic WUM in 2005-2009. case reports. New Medicine. 2009;13(4):85-88.
Rance A, Mittaine M, Michelet M, Martin Blondel A, Labouret G. Delayed diagnosis of foreign body aspiration in children. Arch Pediatrician. 2022 Aug;29(6):424-428. doi: 10.1016/j.arcped.2022.05.006. Epub 2022 Jun 12.
Thank your for your suggestion, we have read the papers with great interest. We have also added the work of Gvetadze et al into the references and main text whereas the work of Rance et al was already referenced in our paper.
We have also corrected the language and editing.
Best regards.
Reviewer 2 Report (New Reviewer)
The manuscript “Foreign body aspiration in children – retrospective study and management novelties” describes a cohort of children aged 0-16 years who manifested with sudden onset of cough, dyspnea, and wheezing, in the absence of other lung conditions. The study includes 242 patients over nine years. The authors point out that rigid fibrochoscopy is still the gold-standard in treating children with FBA, albeit the several important local complications such as airway edema, bleeding, bronchospasm, and inherent issues provoked by general anesthesia. One of the points made is that patients from rural as opposed to urban areas were over-represented in the study and the authors conclude that this is because of socio-economic status, lack of supervision of the children and poor education. The study is interesting and provides new insight into how to diagnose and treat foreign body aspiration.
There are some concerns which need to be addressed:
1) On what basis did the authors draw the conclusion about people in rural areas not supervising their children properly, being alcoholics and generally not being educated? Information about how this conclusion was reached must be included.
2) It is unclear whether the criteria described in the discussion were applied to the cohort included. This must be clearly stated.
3) Language editing is required.
The manuscript needs editing when it comes to word choice, use of contracted form and how a formal text is written.
Author Response
Dear Reviewer 2,
Thank your for your suggestions and comments. We have decided to answer and incorporate them as follows:
- On what basis did the authors draw the conclusion about people in rural areas not supervising their children properly, being alcoholics and generally not being educated? Information about how this conclusion was reached must be included.
The conclusion is mostly based on medical files, family history and interaction with parents. Whereas we do agree this is not sufficient to draw such a conclusion, we believed it was worth mentioning.
- It is unclear whether the criteria described in the discussion were applied to the cohort included. This must be clearly stated.
We do agree therefore we added a statement regarding Haller criteria of diagnosis (Lines 246-250) whereas for Fasseeh et al criteria, it was already stated in the manuscript (Lines 266-270).
- 3) Language editing is required.
We have edited, corrected and even replaced the graphics to make it proper. Hopefully it meets the criteria now.
Thank you, best regards.
Round 2
Reviewer 1 Report (New Reviewer)
Accept in present form
This manuscript is a resubmission of an earlier submission. The following is a list of the peer review reports and author responses from that submission.
Round 1
Reviewer 1 Report
Thank you for the opportunity to review your manuscript titled "Foreign body aspiration in children - management novelties". This article presents a study of 242 pediatric patients diagnosed with foreign body aspiration (FBA) in the Emergency Clinical Hospital for Children in IaÅŸi, Romania, between 2010 and 2018. The study found that the highest incidence of FBA was in rural areas and in children aged 1-3 years, with coughing and dyspnea being the main symptoms leading to hospitalization. The unequal distribution was found to be related to socio-economic status and consumption of inappropriate food. The authors conclude that FBA is a major medical emergency that requires attention and management, and that the challenge remains in identifying asymptomatic or mild symptomatic cases and in managing cases with radiolucent foreign bodies.
The article provides valuable information on the demographics of pediatric patients with foreign body aspiration and the main symptoms leading to hospitalization. The findings highlight the need for further research on ways to prevent foreign body aspiration, especially in rural areas and in children from socio-economically disadvantaged backgrounds. The conclusions emphasize the importance of prompt and effective management of FBA, and the ongoing challenges in the field.
This is a well-organized and presented study. I recommend the authors to use Ulas et al.'s study titled 'Foreign body aspirations in children and adults', which is a large case series and published in the american journal of surgery, to enrich their articles. Minor typos can be edited, as the spelling of 'brobronchoscopy' on line 18 should be corrected to 'bronchoscopy'.
Reviewer 2 Report
Thank you for the opportunity to revise the manuscript entitled “Foreign body aspiration in children - management novelties”, which put attention on a major issue for children medicine.
The manuscript need to be improved, especially in the methods section. Data presented are only descriptive in nature, and the authors have made no inference. The main question is: what this work adds to the literature? Why it is necessary to describe a well-known problem of public health?
Title and abstract
The study’s design - with a commonly used term – is missing in the title or the abstract.
Introduction
What is the rationale for the investigation?
State specific objectives here, including any hypotheses.
Methods
This section must be improved. Check the EQUATOR network and adhere to the most appropriate checklist for your study (es. STROBE?).
What is the study design? Please, declare it, describe the steps performed and put a methodological reference.
The authors approached the “Sfanta Maria” Emergency Clinical Hospital 78 for Children, IaÈ™i (Santa?). Can the authors provide more details about the setting?
“We included in the study a group of 242 patients with ages between 0-16 years”. Give the eligibility criteria, and the sources and methods of selection of participants.
Explain how quantitative variables were handled in the analyses.
Also, describe any efforts to address potential sources of bias.
Results
Have the authors the permission to share the chest radiography figure? If so, please declare it.
No correlation have been made by authors. The authors can think about doing more statistical analysis (e.g., foreign body aspiration has a different distribution between males and females? Alternatively, is there a correlation between age and foreign body aspiration?).
These results may enrich the debate in the discussion.
Author Response
Dear Reviewer 2,
We have carefully read and considered your observations and decided to answer as follows:
- Title and abstract. The study’s design - with a commonly used term – is missing in the title or the abstract.
We agree with this therefore we have added in the title – retrospective study, as well as mentioned in the abstract that it is a retrospective study over the span of 9 years.
- What is the rationale for the investigation? State specific objectives here, including any hypotheses.
We have stated the objectives in the first paragraph of the materials and methods, however we do agree that it should have been in the introduction instead as is preparing and explaining to the reader the purpose and main points of the paper (Lines 71-77).
- This section must be improved. Check the EQUATOR network and adhere to the most appropriate checklist for your study (es. STROBE?).
STROBE Statement—checklist of items that should be included in reports of observational studies
|
|
Item No |
Recommendation |
Page |
||
|
Title and abstract |
1 |
(a) Indicate the study’s design with a commonly used term in the title or the abstract |
1 |
||
|
(b) Provide in the abstract an informative and balanced summary of what was done and what was found |
1 |
||||
|
Introduction |
|||||
|
Background/rationale |
2 |
Explain the scientific background and rationale for the investigation being reported |
1-2 |
||
|
Objectives |
3 |
State specific objectives, including any prespecified hypotheses |
2 |
||
|
Methods |
|||||
|
Study design |
4 |
Present key elements of study design early in the paper |
2 |
||
|
Setting |
5 |
Describe the setting, locations, and relevant dates, including periods of recruitment, exposure, follow-up, and data collection |
2 |
||
|
Participants |
6 |
(a) Cohort study—Give the eligibility criteria, and the sources and methods of selection of participants. Describe methods of follow-up Case-control study—Give the eligibility criteria, and the sources and methods of case ascertainment and control selection. Give the rationale for the choice of cases and controls Cross-sectional study—Give the eligibility criteria, and the sources and methods of selection of participants |
2 |
||
|
(b) Cohort study—For matched studies, give matching criteria and number of exposed and unexposed Case-control study—For matched studies, give matching criteria and the number of controls per case |
|
||||
|
Variables |
7 |
Clearly define all outcomes, exposures, predictors, potential confounders, and effect modifiers. Give diagnostic criteria, if applicable |
3 |
||
|
Data sources/ measurement |
8* |
For each variable of interest, give sources of data and details of methods of assessment (measurement). Describe comparability of assessment methods if there is more than one group |
2-3 |
||
|
Bias |
9 |
Describe any efforts to address potential sources of bias |
- |
||
|
Study size |
10 |
Explain how the study size was arrived at |
2 |
||
|
Quantitative variables |
11 |
Explain how quantitative variables were handled in the analyses. If applicable, describe which groupings were chosen and why |
3 |
||
|
Statistical methods |
12 |
(a) Describe all statistical methods, including those used to control for confounding |
3 |
||
|
(b) Describe any methods used to examine subgroups and interactions |
|
||||
|
(c) Explain how missing data were addressed |
|
||||
|
(d) Cohort study—If applicable, explain how loss to follow-up was addressed Case-control study—If applicable, explain how matching of cases and controls was addressed Cross-sectional study—If applicable, describe analytical methods taking account of sampling strategy |
|
||||
|
(e) Describe any sensitivity analyses |
|
||||
|
Results |
|||||
|
Participants |
13* |
(a) Report numbers of individuals at each stage of study—eg numbers potentially eligible, examined for eligibility, confirmed eligible, included in the study, completing follow-up, and analysed |
3 |
||
|
(b) Give reasons for non-participation at each stage |
|
||||
|
(c) Consider use of a flow diagram |
|
||||
|
Descriptive data |
14* |
(a) Give characteristics of study participants (eg demographic, clinical, social) and information on exposures and potential confounders |
3-4 |
||
|
(b) Indicate number of participants with missing data for each variable of interest |
|
||||
|
(c) Cohort study—Summarise follow-up time (eg, average and total amount) |
|
||||
|
Outcome data |
15* |
Cohort study—Report numbers of outcome events or summary measures over time |
3 |
||
|
Case-control study—Report numbers in each exposure category, or summary measures of exposure |
4-5 |
||||
|
Cross-sectional study—Report numbers of outcome events or summary measures |
6 |
||||
|
Main results |
16 |
(a) Give unadjusted estimates and, if applicable, confounder-adjusted estimates and their precision (eg, 95% confidence interval). Make clear which confounders were adjusted for and why they were included |
4-6 |
||
|
(b) Report category boundaries when continuous variables were categorized |
|
||||
|
(c) If relevant, consider translating estimates of relative risk into absolute risk for a meaningful time period |
|
||||
|
Other analyses |
17 |
Report other analyses done—eg analyses of subgroups and interactions, and sensitivity analyses |
5-6 |
||
|
Discussion |
|||||
|
Key results |
18 |
Summarise key results with reference to study objectives |
7 |
||
|
Limitations |
19 |
Discuss limitations of the study, taking into account sources of potential bias or imprecision. Discuss both direction and magnitude of any potential bias |
8 |
||
|
Interpretation |
20 |
Give a cautious overall interpretation of results considering objectives, limitations, multiplicity of analyses, results from similar studies, and other relevant evidence |
7-8 |
||
|
Generalisability |
21 |
Discuss the generalisability (external validity) of the study results |
8 |
||
|
Other information |
|||||
|
Funding |
22 |
Give the source of funding and the role of the funders for the present study and, if applicable, for the original study on which the present article is based |
8 |
||
- What is the study design? Please, declare it, describe the steps performed and put a methodological reference.
We have included this information in the methods section (Lines 86)
- The authors approached the “Sfanta Maria” Emergency Clinical Hospital 78 for Children, IaÈ™i (Santa?). Can the authors provide more details about the setting?
The name Sfanta is indeed translated to Saint, however we prefer to keep the original name. Our hospital is a tertiary one and more info about the setting has been added (Lines 86-91).
- “We included in the study a group of 242 patients with ages between 0-16 years”. Give the eligibility criteria, and the sources and methods of selection of participants.
The eligibility criteria include age, time of hospitalization, bronchoscopy performed and confirmed diagnosis (Lines 87-91).
- Explain how quantitative variables were handled in the analyses.
The required information has been added (Lines 110-112).
- Have the authors the permission to share the chest radiography figure? If so, please declare it.
All legal guardians of the children included in our study have signed an informed consent regarding the use and processing of personal data and agreed to the use of patients medical information, including radiographies for educational and scientific publishing. All personal and medical data of our patients have been depersonalized as to not be able to identify. The study was performed with the approval of the hospital Ethics Committee (no. 36915/20.12.2022).
- No correlation have been made by authors. The authors can think about doing more statistical analysis (e.g., foreign body aspiration has a different distribution between males and females? Alternatively, is there a correlation between age and foreign body aspiration?). These results may enrich the debate in the discussion.
In the present study we only performed a descriptive analysis of the patients’ characteristics. In a future study, a more complex one, we will perform other types of statistical analysis as well.
Thank you for your comments.
Best regards.

Reviewer 3 Report
Thank you for the opportunity to review your manuscript on an intersting and important topic.
Introduction.
You mention that FBA is a major problem with significant morbidity and mortality - can you describe this objectively with references?
"The shape of the foreign body is an indicator for its final destination in the airway with slimmer, sharper objects travelling further down the bronchial tree becoming extremely dangerous if they reach the cricoid membrane" - the cricoid membrane is proximal to the bronchial tree.
"However, in some cases the foreign bodies can migrate to the stomach with larger objects being able to cause pyloric obstruction accompanied by vomiting and eating refusal" - this was difficult to understand, please clarify with a preceeding comment about upper airway, as the previous comments discuss FBA at the tracheal level, which is difficult to migrate to the ailementary tract.
Materials and Methods
Methods do not describe how you will analyse the data
Results
You have some discussion points in your results eg 'The case distribution is explained on one hand by the fact that children between 1-3 years old tend to explore the environment and thus can aspirate different foods or objects, and on the other hand by the lack of adequate parental supervision.' and 'This led to the logical conclusion that part of the prophylaxis of foreign body aspiration is related to educating the population and applying measures destined to increase living standards through appropriate policies.'
From your methods, i cannot see how you can determine 'rural areas, from dysfunctional families, where poverty, lack of health education, divorces, alcohol consumption, and toxic living environment have had their say.' Is this an inference simply because they live in rural areas? If so, there are a lot of confounding factors and is not a scientific statement.
Figure 1. b should be a new figure, and is LEFT not RIGHT lung atelectesis. Fig 1c is a poor image and does not clearly show thoracic emphysema (or did you mean pneumothorax)
Discussion
Im not sure a discussion about scoring systems should feature prominently as it introduces new detail rather than discussing the results and original hypothesis. Unless you are doing a review of scoring systems as it relates to your cohort, it should not be discussed as in depth as it has. The issue here is that you go on to give an opinion about best scoring system without actually going through the process of retrospectively applying the scores to your cohort. If you do choose to discuss scoring systems, it is important that this also feature in the introduction.
Conclusion
As above